# Oncological and functional outcomes of high-risk and very high-risk prostate cancer patients after robot-assisted radical prostatectomy

Wei-Hsin Chen[1,2], Yu Khun Lee[1,2], Hann-Chorng Kuo[1,2], Jen-Hung Wang[3], Yuan-Hong Jiang[1,2]*

1 Department of Urology, Hualien Tzu Chi Hospital, Buddhist Tzu Chi Medical Foundation, Hualien, Taiwan, 2 Department of Urology, School of Medicine, Tzu Chi University, Hualien, Taiwan, 3 Department of Medical Research, Hualien Tzu Chi Hospital, Buddhist Tzu Chi Medical Foundation, Hualien, Taiwan

* redeemerhd@gmail.com

**Data Availability Statement:** All relevant data are within the manuscript and its Supporting information files.

## Abstract

### Purpose

This study investigated the oncological and functional outcomes of robot-assisted radical prostatectomy (RaRP) in high-risk and very high-risk prostate cancer patients.

### Materials and methods

One hundred localized prostate cancer patients receiving RaRP from August 2015 to December 2020 were retrospectively enrolled. According to NCCN risk classification, patients were classified into two groups, below high-risk group, and high-risk/very high-risk group, to analyze continence outcome within postoperative year one and biochemical recurrence-free survival.

### Results

The mean age of the cohort was 69.7 ± 7.4 years with a median follow-up of 26.4 (range 3.3–71.3) months. Among them, 53%, and 47% patients were below high-risk group, and high-risk/very high-risk group, respectively. The median biochemical recurrence-free survival of the entire cohort was 53.1 months. The high-risk/very high-risk group without adjuvant treatment had significantly worse biochemical recurrence-free survival than the high-risk/very high-risk group with adjuvant treatment (19.6 vs. 60.5 months, $p = 0.029$). Rates of postoperative stress urinary incontinence at 1 week, 1 month, and 12 months were 50.7%, 43.7%, and 8.5%, respectively. High-risk/very high-risk patients had significantly higher rates of stress urinary incontinence at postoperative week 1 (75.8% vs. 28.9%) and month 1 (63.6% vs. 26.3%) than the below high-risk group (both $p < 0.01$). Rates of stress urinary incontinence after RaRP did not differ between two groups from postoperative 3 months to 12 months. The factor of high-risk / very high-risk group was a predictor of immediate but not for long-term postoperative stress urinary incontinence.

**Funding:** No funding was received for this research.

**Competing interests:** The authors have no conflicts of interest to declare.

## Conclusions

High-risk and very high-risk prostate cancer patients receiving a combination of RaRP and adjuvant treatment had comparable biochemical recurrence-free survival to below high-risk prostate cancer patients. The high-risk/very high-risk factor impeded early but not long-term postoperative recovery of continence. RaRP can be considered a safe and feasible option for high-risk and very high-risk prostate cancer patients.

## Introduction

Robot-assisted radical prostatectomy (RaRP) is established as the standard curative treatment for low- and intermediate-risk localized prostate cancer in patients with an estimated survival of more than 10 years [1, 2]. Nevertheless, RaRP is optional in high-risk and very high-risk localized prostate cancer patients and multimodal treatment is recommended by the guidelines of the *National Comprehensive Cancer Network®* (NCCN) guidelines and the European Association of Urology [1, 2]. Apart from radical prostatectomy, external beam radiotherapy (EBRT) combined with androgen deprivation therapy (ADT) is also a viable treatment option in high-risk and very high-risk cases. Recent meta-analyses have shown radical prostatectomy to be superior to EBRT with or without ADT in terms of cancer-specific mortality in high-risk and very high-risk patients [3, 4]. Therefore, RaRP has played an increasing role in the treatment of these patients over the past decade [5, 6].

Despite these superior oncological outcomes, RaRP in high-risk and very high-risk prostate cancer patients presents issues due to higher rates of urinary incontinence and erectile dysfunction after radical prostatectomy than radiotherapy [7]. RaRP in high-risk and very high-risk prostate cancer patients also tends to be more complicated because of more advanced tumor progression and the need for extensive pelvic lymph node dissection. Furthermore, continence-maintaining techniques such as bladder neck and apex preservation are not always possible because of the high rate of extraprostatic cancer. Previous studies have reported 1 year continence rates after RaRP ranging from 30% to 100% in high-risk and very high-risk prostate cancer patients [8–10]. Yet, there have been small amount of literature discussing the trifecta outcome of RaRP. Ou et al. reported nearly no difference in continence rates and potency within one year in patients with low-risk, intermediate-risk, and high-risk patient after RaRP [11]. High-risk patients have significantly high rate of biochemical recurrence.

In this study, we aimed to compare the oncological and functional outcomes of RaRP in high-risk and very high-risk localized prostate cancer patients with their low-risk and intermediate-risk counterparts.

## Materials and methods

We retrospectively enrolled one hundred patients with localized prostate cancer who underwent RaRP between August 2015 and December 2020 at a single medical center. All surgeries were performed using a standard transperitoneal approach with both anterior and posterior reconstruction by a single surgeon (YHJ). All patients had a thorough preoperative clinical staging workup, including a digital rectal exam, measurement of initial serum prostate-specific antigen (iPSA), transrectal ultrasound-guided prostate biopsy, a multiparametric magnetic resonance imaging (mpMRI) scan of the prostate, and a whole-body bone scan. All patients were classified as low-risk, intermediate-risk, high-risk or very high-risk using the initial risk stratification of the 2021 NCCN guidelines, which were derived from D'Amico criteria, based

on their preoperative workup. According to the criteria, patients with a serum PSA less than 10 ng/mL, a clinical T staging less than T2a, and grade group 1 prostate cancer were initially stratified into low-risk group. Patients with one of the following features were stratified into high-risk group: T3a prostate cancer, ISUP grade group 4 pathology, or serum PSA higher than 20 ng/mL. Patients with a least one of the following features were stratified into very high-risk group: T3b or T4 prostate cancer, and/or a primary Gleason pattern 5 pathology, and/or having 2 or 3 high-risk features, and/or having more than 4 cores of ISUP grade group 4/5 biopsy. Patients were neither low- or high-/very high-risk were stratified into intermediate-risk group.

The patients were further categorized into two groups: a below high-risk group (<high risk) and a high-risk/very high-risk group (≥high risk). The former included low-risk, and intermediate-risk patients and the latter consisted of high-risk and very high-risk patients. Data on parameters relevant to oncological and functional outcomes were collected. Preoperative parameters included age, total prostate volume (TPV), body mass index (BMI), membranous urethra length (MUL) on mpMRI [12], the use of ADT before RaRP, and the history of transurethral resection of the prostate (TURP) or transurethral incision of the prostate before RaRP. Perioperative parameters included operation time, console time, blood loss, and the preservation of neurovascular bundles (NVBs). Pathological findings included a positive surgical margin, the pathological cancer stage, lymph node involvement, adjuvant radiotherapy, and adjuvant ADT.

All patients received standardized perioperative care and underwent a cystography on postoperative day 6. The urethral catheter was removed the following day if there were no significant postoperative complications or abnormal findings on the cystography. Urinary incontinence status, including stress urinary incontinence and urgency urinary incontinence, was assessed in the first week, the first month, the third month, the sixth month, the ninth month, and the twelfth month after RaRP. The definitions of stress and urgency urinary incontinence were based on International Continence Society criteria [13]. Stress urinary incontinence was defined as involuntary loss of urine resulting from effort, physical exertion, sneezing, or coughing. Urgency urinary incontinence was defined as involuntary loss of urine associated with urgency.

All the patients were regularly followed up at clinic after discharge. The intervals of postoperative clinic visits were monthly within the first three months followed by every three months, with additional visits if clinically indicated. Patients with adverse clinical findings, including a positive surgical margin, persistently elevated serum PSA, and extraprostatic and/or lymph node involvement were referred for adjuvant radiation therapy with or without ADT 2–6 months after RaRP. Patients' serum PSA levels were measured regularly during follow-up for biochemical recurrence. Biochemical recurrence was defined as the elevation of serum PSA level above 0.2 ng/mL after RaRP with or without adjuvant therapy.

The primary endpoint of each patient's oncological outcome was biochemical recurrence-free survival, and the primary endpoint of their functional outcome was the rate of stress urinary incontinence and urgency urinary incontinence within 1 year of RaRP. These were compared between the below high-risk group and the high-risk/very high-risk group. The high-risk/very high-risk group patients were divided into two subgroups according to whether or not they received adjuvant treatment. The secondary endpoint of the functional outcome was the identification of predictive factors for postoperative stress urinary incontinence and urgency urinary incontinence.

This study was approved by the Institutional Review Board and Ethics Committee of Buddhist Tzu Chi General Hospital (IRB107-38-A). Each patient was informed about the study rationale and procedures, and written informed consent was obtained. All methods were performed in accordance with the relevant guidelines and regulations.

## Statistical analysis

Biochemical recurrence-free survival was calculated with Kaplan–Meier survival analysis. *P*-values were determined via Log-rank test. The means of continuous data were compared with a t-test and/or a one-way analysis of variance. The medians of continuous data were compared with a two-sided Mann–Whitney *U*-test and/or a Kruskal–Wallis test. Categorical parameters were analyzed with chi-square tests. Predictive factors for postoperative stress urinary incontinence and urgency urinary incontinence were determined via univariable and multivariable logistic regression. The continuous data was reported as mean ± SD (standard deviation) and median (range). All calculations were performed using SPSS for MacOS Big Sur, version 25.0 (IBM Corp., Armonk, N.Y., USA). *P*-values of <0.05 were considered significant.

## Results

One hundred male patients with a mean age of 69.7 ± 7.4 years were followed up for a median of 26.4 (range 3.3–71.3) months. Of these 100, 53% and 47% were in the below high-risk group (<high risk) and the high-risk/very high-risk group (≥high risk), respectively (Table 1). Of the below high-risk patients, 20 and 33 patients were low- and intermediate-risk, respectively. Eleven and 36 patients were high-risk and very high-risk, respectively (S1 Table in S1 File).

### Patient demography

The high-risk/very high-risk group had an older average age at the time of surgery, a larger TPV, a higher iPSA, and a higher biopsy Gleason score (a measure of the amount of abnormal tissue) than the below high-risk group (Table 1). A significantly higher percentage of the high-risk/very high-risk group than the below high-risk group received preoperative ADT. Perioperative parameters, including overall operation time, console time, and blood loss, did not differ significantly between the two groups, except for neurovascular bundle preservation, of which there was a lower proportion in the high-risk/very high-risk group. For the pathological parameters, the high-risk/very high-risk group had significantly higher pathological T-staging (tumor size), higher pathological Gleason scores (pGleason score), a higher proportion of lymph node involvement, and a higher percentage of positive surgical margins than the below high-risk group. However, below high-risk group patients had a significantly higher percentage of upstaging (45.3%) than the high-risk/very high-risk group patients (34.0%). Additionally, significantly more high-risk/very high-risk group patients received adjuvant radiotherapy and adjuvant ADT. The follow-up duration was similar between the two groups.

### Oncological outcomes

Overall, 36 (36%) patients developed biochemical recurrence with a median time to biochemical recurrence of 16.2 (range 3.5–60.5) months. The median biochemical recurrence-free survival of the entire cohort was 53.1 months. As shown in Fig 1A, high-risk/very high-risk patients had significantly shorter biochemical recurrence-free survival than below high-risk patients (median 34.7 vs. 53.1 months, *p* = 0.04). High-risk/very high-risk patients without adjuvant treatment (n = 21) had significantly shorter biochemical recurrence-free survival than those with adjuvant treatment (n = 26) (median 19.6 vs. 60.5 months, *p* = 0.029) (Fig 1B). The median biochemical recurrence-free survival of the high-risk/very high-risk group with adjuvant treatment was comparable with that of the below high-risk group (n = 53) (median 60.5 vs. 53.1 months, *p* = 0.69).

**Table 1. Preoperative, perioperative, pathological, and postoperative characteristics of prostate cancer patients (the entire study cohort, n = 100).**

| | | Total | Risk group | | | | p-value |
|---|---|---|---|---|---|---|---|
| | | | <High Risk | | ≥High Risk | | |
| | | | Low | Intermediate | High | Very high | |
| Numbers | | 100 (100.00%) | 53 (53.00%) | | 47 (47.00%) | | |
| | | | 20 (20.00%) | 33 (33.00%) | 11 (11.00%) | 36 (36.00%) | |
| **Preoperative** | | | | | | | |
| Age | | 69.7 ± 7.4 | 67.9 ± 7.5 | | 71.6 ± 6.9 | | **0.011** |
| TPV (mL) | | 36.3 ± 14.6 | 33.3 ± 11.9 | | 39.7 ± 16.7 | | **0.028** |
| iPSA, median (IQR) (ng/mL) | | 12.8 (7.3–25.7) | 8.1 (5.7–11.7) | | 26.5 (15.3–49.8) | | **<0.001** |
| bGleason score | 6 | 58 (58.00%) | 40 (75.50%) | | 18 (38.30%) | | **<0.001** |
| | 7 | 27 (27.00%) | 13 (24.50%) | | 14 (29.80%) | | |
| | ≥8 | 15 (15.00%) | 0 (0.00%) | | 15 (31.90%) | | |
| BMI (kg/m$^2$) | | 25.9 ± 3.2 | 25.9 ± 3.0 | | 25.9 ± 3.5 | | 0.966 |
| MUL (mm) | | 12.5 ± 2.5 | 12.6 ± 2.5 | | 12.4 ± 2.4 | | 0.567 |
| Preoperative ADT | | 11 (11.00%) | 2 (3.80%) | | 9 (19.10%) | | **0.014** |
| Preoperative TURP/TUIP | | 24 (24.00%) | 13 (24.50%) | | 11 (23.40%) | | 0.895 |
| **Perioperative** | | | | | | | |
| Overall operative time (min) | | 191.6 ± 30.5 | 189.8 ± 27.6 | | 193.7 ± 33.6 | | 0.521 |
| Console time (min) | | 140.6 ± 33.6 | 139.2 ± 31.9 | | 142.1 ± 35.8 | | 0.673 |
| Blood loss (mL) | | 107.6 ± 79.9 | 110.2 ± 72.1 | | 104.8 ± 88.4 | | 0.877 |
| NVB preservation | | 43 (43.00%) | 37 (69.80%) | | 6 (12.80%) | | **<0.001** |
| **Pathological results** | | | | | | | |
| pT stage | pT2 | 58 (58.00%) | 45 (84.90%) | | 13 (27.70%) | | **<0.001** |
| | pT3 | 40 (40.00%) | 8 (15.10%) | | 32 (68.10%) | | |
| | pT4 | 2 (2.00%) | 0 (0.00%) | | 2 (4.30%) | | |
| Pathological Gleason score | 6 | 41 (41.00%) | 28 (52.80%) | | 13 (27.70%) | | **<0.001** |
| | 7 | 41 (41.00%) | 22 (41.50%) | | 19 (40.40%) | | |
| | ≥8 | 18 (18.00%) | 3 (5.70%) | | 15 (31.90%) | | |
| Lymph node involvement | | 7 (7.10%) | 0 (0.00%) | | 7 (15.20%) | | **0.003** |
| Positive surgical margin | | 18 (18.00%) | 3 (5.70%) | | 15 (31.90%) | | **<0.001** |
| Upstaging | | 40 (40.00%) | 24 (45.30%) | | 16 (34.00%) | | **0.023** |
| Upgrading | | 32 (32.00%) | 17 (32.10%) | | 15 (31.90%) | | 0.508 |
| **Postoperative** | | | | | | | |
| Adjuvant radiotherapy | | 30 (30.00%) | 7 (13.20%) | | 23 (48.90%) | | **<0.001** |
| Adjuvant ADT | | 21 (21.00%) | 4 (7.50%) | | 17 (36.20%) | | **<0.001** |
| Follow-up duration, median (range) (months) | | 26.4 (3.3–71.3) | 27.5 (3.3–64.2) | | 24.0 (4.3–71.3) | | 0.689 |

ADT, androgen deprivation therapy; bGleason score, biopsy Gleason score; BMI, body mass index; iPSA, initial serum prostate-specific antigen; IQR, interquartile range; MUL, membranous urethra length; NVB, neurovascular bundles; pGleason score, pathological Gleason score; pT staging, pathological T staging; TPV, total prostate volume; TUIP, transurethral incision of the prostate; TURP, transurethral resection of the prostate.

## Functional outcomes

Seventy-one patients (71%) in our cohort had documented function outcomes within one year after RaRP. This included the incidence of postoperative stress urinary incontinence and urgency urinary incontinence. The mean age was 69.8 ± 7.7 years, and there was no significant difference in age between groups. Of these 71 patients, 38 (53.5%) were in the below high-risk group and 33 (46.5%) were in the high-risk/very high-risk group (S2 Table in S1 File). Among

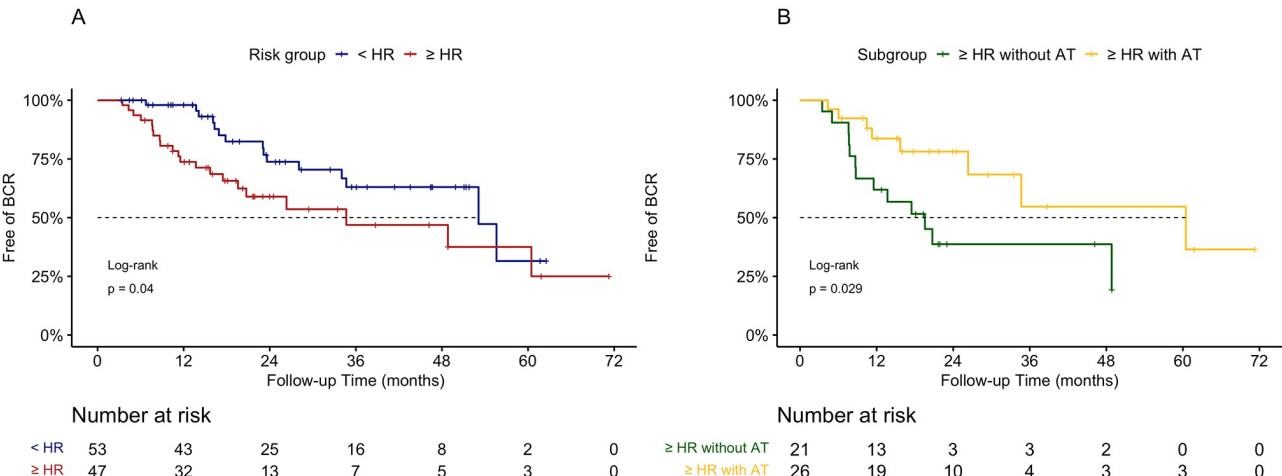

**Fig 1.** A: Biochemical recurrence (BCR)-free survival among prostate cancer patients in the below high-risk group (<HR) and the high-risk/very high-risk group (≥HR). B: BCR-free survival among patients in the high-risk/very high-risk group, without adjuvant treatment (≥HR without AT), and high-risk/very high-risk group, with adjuvant treatment (≥HR with AT).

the 71 patients, high-risk/very high-risk patients had significantly larger total prostate volume, a significantly higher proportion of receiving preoperative ADT, less percentage of neurovascular bundle preservation, and higher proportion of receiving adjuvant radiotherapy and ADT. The rate of stress urinary incontinence trended down from 50.7% at postoperative week 1 (just after removal of the urethral catheter) to 8.5% at postoperative month 12 (Fig 2A). The rates of stress urinary incontinence at both postoperative week 1 and postoperative month 1 were significantly higher in the high-risk/very high-risk patients than in the below high-risk patients (75.8% vs. 28.9%, $p < 0.001$; 63.6% vs. 26.3%, $p = 0.002$). The rate of stress urinary incontinence did not differ between 3 and 12 months, postoperatively. The rates of urgency

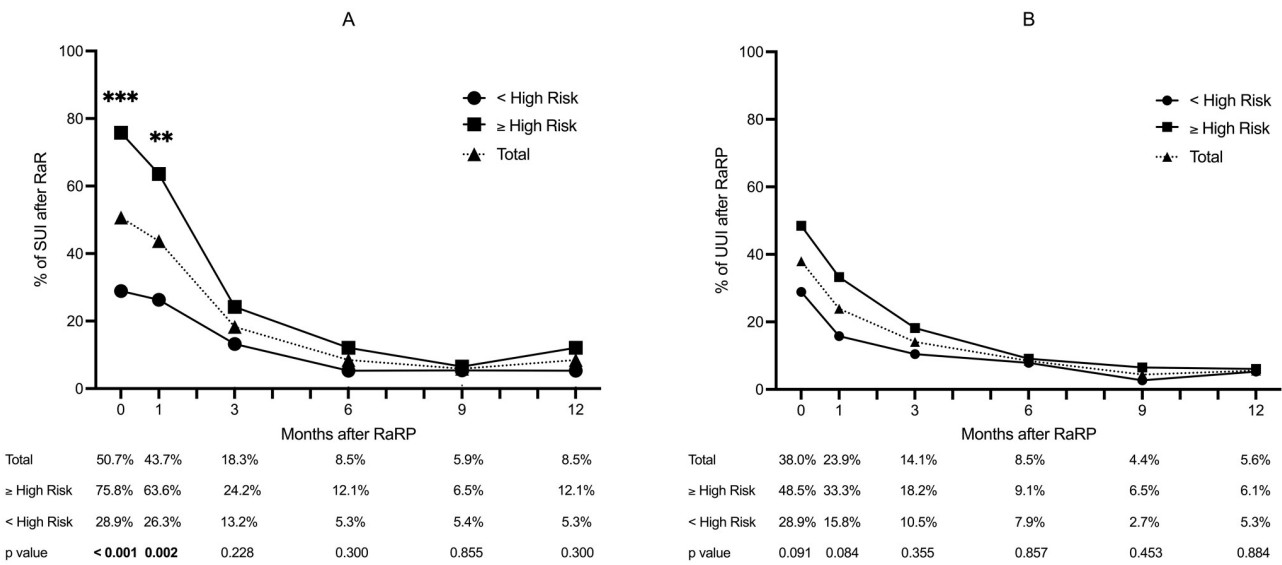

**Fig 2.** A: Proportion of prostate cancer patients with stress urinary incontinence (SUI) after robot-assisted radical prostatectomy over time. B: Proportion of patients with urgency urinary incontinence (UUI) after robot-assisted radical prostatectomy over time. $^*p < 0.05$, $^{**}p < 0.01$, $^{***}p < 0.001$.

urinary incontinence also trended down over time but there were no significant differences between the below high-risk group and high-risk/very high-risk group (Fig 2B).

Table 2 shows the univariable and multivariable logistic regression analyses of the risk of stress urinary incontinence in our patients at different time points after RaRP. High biopsy Gleason scores ($\geq$8) (odds ratio [OR] 7.59, confidence interval [CI] 1.46–39.63, $p$ = 0.016), high-risk/very high-risk (OR 7.67, CI 2.66–22.16, $p$ < 0.001), NVB preservation (OR 0.36, CI 0.14–0.97, $p$ = 0.044), high pathological T staging (pT $\geq$ 3) (OR 5.30, CI 1.88–14.93, $p$ = 0.002), and high pGleason scores ($\geq$8) (OR 7.50, CI 1.38–40.69, $p$ = 0.020) were predictive factors for stress urinary incontinence at postoperative week 1 in the univariable logistic regression model. High-risk/very high-risk (OR 5.33, CI 1.22–23.31, $p$ = 0.026) was the only independent predictor of stress urinary incontinence at postoperative week 1 in the multivariable logistic regression model. A larger total prostate volume (OR 1.06, CI 1.01–1.11, $p$ = 0.016), a high biopsy Gleason score ($\geq$8) (OR 15.75, CI 1.45–171.22, $p$ = 0.024), and a positive surgical margin (OR 16.57, CI 2.56–107.50, $p$ = 0.003) were predictive of stress urinary incontinence at postoperative month 12 in the univariable logistic regression model. No significant predictors of stress urinary incontinence at either postoperative month 6 or 12 were identified using multivariable logistic regression.

S3 Table in S1 File shows the logistic regression models of the risks of urgency urinary incontinence at different time points after RaRP. A shorter membranous urethra length (OR 0.78, CI 0.62–0.99, $p$ = 0.038) was the only predictive factor of urgency urinary incontinence at postoperative week 1 in the univariable logistic regression model. No significant predictive factors for urgency urinary incontinence were identified at postoperative 6 and 12 month.

## Discussion

In this study, we determined biochemical recurrence-free survival rates and the incidence of postoperative urinary incontinence in high-risk/very high-risk and below high-risk localized prostate cancer patients after RaRP. We found that high-risk and very high-risk patients had shorter biochemical recurrence-free survival than below high-risk prostate cancer patients. The high-risk/very high-risk group without adjuvant treatment had significantly worse biochemical recurrence-free survival than the high-risk/very high-risk group with adjuvant treatment. The rate of stress urinary incontinence trended down from 50.7% at postoperative week one (immediately after removal of the urethral catheter) to 8.5% at postoperative month 12. Compared with the below high-risk group, the high-risk/very high-risk group had significantly higher stress urinary incontinence rates at postoperative week 1 and month 1 and comparable stress urinary incontinence rates at postoperative month 12. Additionally, high-risk/very high-risk was a significant predictor for immediate postoperative stress urinary incontinence but not for long-term stress urinary incontinence in the multivariable logistic regression model. Considering the oncological and functional outcomes, RaRP can be considered a safe and feasible option for high-risk and very high-risk prostate cancer patients.

High-risk/very high-risk prostate cancer patients had a worse biochemical recurrence-free rate than their low- and intermediate-risk counterparts after radical prostatectomy. Abdollah et al. have reported a 10-year biochemical recurrence-free rate of 70% in low-risk patients, 36% in intermediate-risk patients, 31% in high-risk patients, and 26% in very high-risk patients after radical prostatectomy [14]. The overall biochemical recurrence rate in high-risk patients with locally advanced prostate cancer ranged from 13% to 35% and 18.5% to 28.6%, respectively [5, 6]. Koo et al. found that 77% of high-risk and 58% of very high-risk patients were free from biochemical recurrence at final follow-up, with median follow-up durations of 31.1 and 36.1 months, respectively [9]. Gandaglia et al. reported a 3-year biochemical

**Table 2. Univariable and multivariable logistic regression models of risks of stress urinary incontinence in prostate cancer patients with functional documentation (n = 71) 1 week, 6 months, and 12 months after robot-assisted radical prostatectomy.**

| | | Stress Urinary Incontinence at 1 Week* | | | | Stress Urinary Incontinence at 6 Months | | | | Stress Urinary Incontinence at 12 Months | | | |
| --- | --- | --- | --- | --- | --- | --- | --- | --- | --- | --- | --- | --- | --- |
| | | Crude | | Adjusted | | Crude | | Adjusted | | Crude | | Adjusted | |
| | | OR [95% CI] | p-value | OR [95% CI] | p-value | OR [95% CI] | p-value | OR [95% CI] | p-value | OR [95% CI] | p-value | OR [95% CI] | p-value |
| Preoperative | | | | | | | | | | | | | |
| Age at OP | | 0.99 [0.93–1.05] | 0.736 | 0.95 [0.88–1.03] | 0.232 | 0.94 [0.84–1.06] | 0.310 | – | – | 0.99 [0.89–1.10] | 0.846 | 0.97 [0.84–1.12] | 0.653 |
| TPV | | 1.03 [0.99–1.06] | 0.154 | – | – | 1.04 [1.00–1.09] | 0.074 | – | – | 1.06 [1.01–1.11] | **0.016** | 1.04 [0.98–1.11] | 0.193 |
| iPSA | | 1.01 [0.99–1.02] | 0.468 | – | – | 0.99 [0.95–1.03] | 0.496 | – | – | 1.01 [0.99–1.03] | 0.336 | – | – |
| bGleason score | 6 | Ref | Ref | Ref | Ref | Ref | Ref | – | – | Ref | Ref | Ref | Ref |
| | 7 | 3.09 [0.96–9.98] | 0.059 | 2.02 [0.46–8.82] | 0.351 | 12.92 [1.33–126.08] | **0.028** | – | – | 5.60 [0.47–66.32] | 0.172 | 3.69 [0.26–52.82] | 0.337 |
| | ≥8 | 7.59 [1.46–39.63] | **0.016** | 1.30 [0.16–10.64] | 0.807 | 4.20 [0.24–73.07] | 0.325 | – | – | 15.75 [1.45–171.22] | **0.024** | 6.17 [0.38–100.17] | 0.201 |
| BMI | | 1.00 [0.87–1.16] | 0.974 | – | – | 1.10 [0.85–1.43] | 0.451 | – | – | 1.05 [0.81–1.35] | 0.712 | - | – |
| MUL | | 0.90 [0.73–1.11] | 0.315 | – | – | 1.13 [0.78–1.62] | 0.516 | – | – | 0.86 [0.56–1.32] | 0.491 | - | – |
| Risk group | < High Risk | Ref | Ref | Ref | Ref | Ref | Ref | – | – | Ref | Ref | – | – |
| | ≥ High Risk | 7.67 [2.66–22.16] | **<0.001** | 5.33 [1.22–23.31] | **0.026** | 2.48 [0.42–14.52] | 0.313 | – | – | 2.48 [0.42–14.52] | 0.313 | – | – |
| Preoperative ADT | | 4.71 [0.92–24.04] | 0.062 | – | – | 1.24 [0.13–11.92] | 0.850 | – | – | 0.00 [0.00–10.00] | 0.999 | - | – |
| Preoperative TURP/TUIP | | 0.35 [0.11–1.15] | 0.084 | – | – | 0.67 [0.07–6.16] | 0.721 | – | – | 0.67 [0.07–6.16] | 0.721 | - | – |
| Perioperative | | | | | | | | | | | | | |
| Overall operative time | | 0.99 [0.97–1.01] | 0.229 | – | – | 1.02 [0.99–1.04] | 0.315 | – | – | 1.01 [0.98–1.03] | 0.693 | - | – |
| Console time | | 0.99 [0.97–1.00] | 0.059 | – | – | 1.02 [0.99–1.04] | 0.214 | – | – | 1.02 [0.99–1.04] | 0.272 | – | – |
| Blood loss | | 1.00 [0.99–1.01] | 0.967 | – | – | 1.01 [1.00–1.02] | 0.065 | – | – | 1.01 [1.00–1.02] | 0.370 | – | – |
| NVB preservation | | 0.36 [0.14–0.97] | **0.044** | 0.92 [0.24–3.60] | 0.906 | 0.28 [0.03–2.55] | 0.259 | – | – | 0.00 [0.00–10.00] | 0.998 | – | – |
| Pathological results | | | | | | | | | | | | | |
| pT stage | 2 | Ref | Ref | Ref | Ref | Ref | Ref | – | – | Ref | Ref | – | – |
| | ≥ 3 | 5.30 [1.88–14.93] | **0.002** | 2.34 [0.61–8.93] | 0.214 | 1.41 [0.26–7.51] | 0.689 | – | – | 3.00 [0.51–17.59] | 0.223 | – | – |
| pGleason score | 6 | Ref | Ref | Ref | Ref | Ref | Ref | – | – | Ref | Ref | – | – |
| | 7 | 1.92 [0.69–5.39] | 0.214 | 1.45 [0.39–5.33] | 0.579 | 1.80 [0.28–11.64] | 0.537 | – | – | 3.72 [0.36–37.99] | 0.268 | – | – |
| | ≥8 | 7.50 [1.38–40.69] | **0.020** | 1.81 [0.21–15.89] | 0.593 | 1.50 [0.12–18.36] | 0.751 | – | – | 6.89 [0.56–84.98] | 0.132 | – | – |
| Lymph node involvement | | 1.50 [0.24–9.57] | 0.668 | – | – | 0.00 [0.00–10.00] | 0.999 | – | – | 0.00 [0.00–10.00] | 0.999 | – | – |
| Positive margin | | 3.05 [0.74–12.62] | 0.124 | – | – | 1.10 [0.12–10.44] | 0.934 | – | – | 16.57 [2.56–107.50] | **0.003** | 5.34 [0.53–53.50] | 0.154 |
| Postoperative | | | | | | | | | | | | | |
| Adjuvant radiotherapy | | NA | NA | NA | NA | 3.06 [0.56–16.71] | 0.196 | – | – | 1.41 [0.24–8.42] | 0.705 | – | – |

(*Continued*)

**Table 2.** (Continued)

| | Stress Urinary Incontinence at 1 Week* | | | | Stress Urinary Incontinence at 6 Months | | | | Stress Urinary Incontinence at 12 Months | | | |
|---|---|---|---|---|---|---|---|---|---|---|---|---|
| | Crude | | Adjusted | | Crude | | Adjusted | | Crude | | Adjusted | |
| | OR [95% CI] | p-value | OR [95% CI] | p-value | OR [95% CI] | p-value | OR [95% CI] | p-value | OR [95% CI] | p-value | OR [95% CI] | p-value |
| Adjuvant ADT | NA | NA | NA | NA | 2.21 [0.36–13.49] | 0.391 | – | – | 0.80 [0.09–7.45] | 0.845 | – | – |

ADT, androgen deprivation therapy; bGleason score, biopsy Gleason score; BMI, body mass index; iPSA, initial serum prostate-specific antigen; <high-risk, below high-risk group; ≥high-risk, high-risk/very high-risk group; MUL, membranous urethra length; NA, not available; NVB, neurovascular bundles; OR, odds ratio; pGleason score, pathological Gleason score; pT staging, pathological T staging; RaRP, robot-assisted radical prostatectomy; STRESS URINARY INCONTINENCE, TPV, total prostate volume; TUIP, transurethral incision of the prostate; TURP, transurethral resection of the prostate.

* One week after RaRP, immediately after removal of the urethral catheter

recurrence-free survival rate of 63.3% in patients with at least cT3 localized prostate cancer [10]. Shin et al. reported a 3-year biochemical recurrence-free survival of 78.7% in D'Amico high-risk patients [15]. In our study, the 3-year biochemical recurrence-free rate in high-risk/very high-risk patients was 46.9%, which was lower than that seen in the aforementioned studies. There are several possible reasons for this. In our study, we performed RaRP and standard pelvic lymph node dissection and the enrolled high-risk/very high-risk patients had a higher median PSA level (26.5 ng/mL). Nevertheless, Gandaglia et al. reported data from RaRP and extended pelvic lymph node dissection in locally advanced prostate cancer patients with a median PSA level of 9.7 ng/mL. Similarly, Shin et al. reported data from a younger population (average age 62.5 years), nearly 90% of whom received extended pelvic lymph node dissection. More severe oncological factors (indicating greater tumor burden and involvement) and less extensive lymph node dissection might explain the lower biochemical recurrence-free survival rate in our study.

Adjuvant radiotherapy plays an important but controversial role in biochemical recurrence in high-risk/very high-risk prostate cancer patients who undergo RaRP. Thompson et al. have reported an average of 10.3 years of PSA-relapse-free survival in patients who received radical prostatectomy followed by adjuvant radiotherapy but only 3.1 years in those without adjuvant therapy [16]. In a retrospective review of 26118 localized prostate cancer patients, adjuvant radiotherapy was associated with significantly lower all-cause mortality than early salvage therapy in patients with adverse pathology (pN1, pGleason scores of 8–10, pT3/4) [17]. By contrast, several randomized controlled trials in 2020, including the phase III RADICALS-RT trial and GETUG-AFU 17 trial, found that adjuvant radiotherapy had no significant benefits to survival but led to a higher rate and severity of long-term urinary incontinence compared with early salvage radiotherapy [18, 19]. In our study, high-risk/very high-risk patients who underwent RaRP had similar operation times and blood loss to the below high-risk group. Additionally, high-risk/very high-risk patients who received adjuvant treatment (including radiotherapy and/or ADT) after RaRP had significantly better biochemical recurrence-free survival than those who did not, with biochemical recurrence-free survival comparable with that of below high-risk patients. This result favors the selection of RaRP as an initial step in a multimodal treatment strategy (followed by adjuvant or salvage treatment) in high-risk and very high-risk prostate cancer patients and suggests that this is a safe approach with improved oncological outcomes.

Functional urinary incontinence outcomes reported in high-risk/very high-risk patients who undergo RaRP have varied between studies. Casey et al. have reported rates of 85% and 100% continence at postoperative months 6 and 12 in pT3 prostate cancer patients after RaRP

[8]. Koo et al. have reported 56% and 32% continence rates at postoperative month 12 in high-risk and very high-risk patients, respectively [9]. Gandaglia et al. reported a 64% near-continence rate 12 months after RaRP, with no or one pad per day, in patients with locally advanced prostate cancer [10]. Shin et al. reported 88.3% continence rate at 12 months after RaRP [15]. Varied definitions of continence among these studies may explain the inconsistent continence rates. In this study, the definition of stress urinary incontinence was based on the criteria of the International Continence Society, and the overall stress urinary incontinence rate at postoperative month 12 was 8.5%, which is in accord with the general level [20]. There was no significant difference between the below high-risk group and high-risk/very high-risk group patients. Our results suggest that RaRP is a feasible option in high-risk and very high-risk localized prostate cancer patients that does not impede the recovery of long-term continence.

The continence mechanism in men is controlled by the prostate gland, urethral sphincteric system, and urethral supportive system [21]. After radical prostatectomy, the remaining functions of the urethral sphincter and supportive system play important roles in the recovery of urinary continence. However, maximal preservation of the urethral sphincter and the supportive system cannot always be achieved in high-risk and very high-risk patients because of the high risk of extraprostatic expansion and the need for pelvic lymph node dissection. In this study, high-risk/very high-risk was the only independent predictor of stress urinary incontinence at postoperative week 1 but was not predictive at postoperative month 12 in multivariable logistic regression. The advanced intraoperative and precise dissection techniques used in robotic surgery might decrease the impact of reduced preservation of the urethral sphincter and supportive system. Thus, high-risk/very high-risk only affected early recovery, not long-term recovery, of urinary incontinence.

Previous studies have identified several biological and preoperative predictors of urinary incontinence after radical prostatectomy, including older age [22, 23], high BMI [24], comorbidities [22], increased prostate volume [25], pre-existing lower urinary tract symptoms [24], bladder and sphincter dysfunction [26], and a history of TURP and radiotherapy [27]. Perioperative predictors that have been identified include a less experienced surgeon [28]; damage to, or non-nerve sparing of, the neurovascular bundle [29]; excessive surgical dissection [27]; and adjuvant/salvage radiotherapy [30]. In this study, a larger total prostate volume, a high biopsy Gleason score (≥8), and a positive surgical margin were found using univariable logistic regression to be predictive of stress urinary incontinence at postoperative month 12. No significant predictive factors for either stress urinary incontinence at postoperative month 6 or 12 were identified using the multivariable logistic regression model. Postoperative stress urinary incontinence is a complex summary display affected by the above preoperative and perioperative factors. With the advancement of surgical techniques, the effects of these factors may be expected to lessen.

## Limitations

This study had several limitations. As it was not a randomized trial, there may have been selection biases that would prevent our patients from being a representative sample of this demographic. There might also have been an attrition bias due to a disproportionate loss of patients with better oncological outcomes during follow-up. This was also a retrospective analysis and therefore subjective to potential observational biases. Only 71% of our cohort had documented functional outcomes. However, the proportions of below high-risk and high-risk/very high-risk patients with documented functional outcomes were similar to those for the entire cohort. Finally, the sample size of this study was relatively small, potentially reducing the validity and reliability of our results.

## Conclusion

High-risk and very high-risk prostate cancer patients receiving a combination of RaRP and adjuvant treatment had comparable biochemical recurrence-free survival to below high-risk prostate cancer patients. The high-risk/very high-risk factor impeded early but not long-term recovery of continence after RaRP. Considering the oncological and functional outcomes, RaRP can be considered a safe and feasible option for high-risk and very high-risk prostate cancer patients.

## Supporting information

**S1 File.**
(DOCX)

## Acknowledgments

**Ethics statement:** This study was approved by the Institutional Review Board and Ethics Committee of Buddhist Tzu Chi General Hospital (IRB107-38-A).

## Author Contributions

**Conceptualization:** Yuan-Hong Jiang.

**Formal analysis:** Wei-Hsin Chen, Jen-Hung Wang.

**Methodology:** Jen-Hung Wang, Yuan-Hong Jiang.

**Project administration:** Yuan-Hong Jiang.

**Resources:** Yuan-Hong Jiang.

**Software:** Wei-Hsin Chen, Jen-Hung Wang.

**Supervision:** Yu Khun Lee, Hann-Chorng Kuo, Yuan-Hong Jiang.

**Validation:** Hann-Chorng Kuo.

**Visualization:** Wei-Hsin Chen.

**Writing – original draft:** Wei-Hsin Chen.

**Writing – review & editing:** Yuan-Hong Jiang.

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
