## [Decision Letter · Decision Letter 0]

27 Dec 2022

PONE-D-22-25140Oncological and Functional Outcomes of High-Risk and Very High-Risk Prostate Cancer Patients after Robot-Assisted Radical ProstatectomyPLOS ONE

Dear Dr. Jiang,

Thank you for submitting your manuscript to PLOS ONE. After careful consideration, we feel that it has merit but does not fully meet PLOS ONE’s publication criteria as it currently stands. Therefore, we invite you to submit a revised version of the manuscript that addresses the points raised during the review process.

We look forward to receiving your revised manuscript.

Kind regards,

Mohammad-Mahdi Rashidi, M.D.

Guest Editor

PLOS ONE

Journal Requirements:

"No funding was received for this research."

Reviewers' comments:

Reviewer's Responses to Questions

**Comments to the Author**

1. Is the manuscript technically sound, and do the data support the conclusions?

Reviewer #1: Partly

Reviewer #2: Yes

Reviewer #3: Yes

2. Has the statistical analysis been performed appropriately and rigorously? 

Reviewer #1: Yes

Reviewer #2: Yes

Reviewer #3: Yes

3. Have the authors made all data underlying the findings in their manuscript fully available?

Reviewer #1: No

Reviewer #2: Yes

Reviewer #3: Yes

4. Is the manuscript presented in an intelligible fashion and written in standard English?

Reviewer #1: Yes

Reviewer #2: Yes

Reviewer #3: Yes

5. Review Comments to the Author

Reviewer #1: I would like to thank the editor for this opportunity and the authors foe their efforts. Investigating the outcomes of robot-assisted radical prostatectomy outcomes based on a risk stratification method on included sample of patients and also including the impact of neoadjuvant therapy besides the surgery in this study is interesting and adds to the literature. Although the study notion is not novel, the methodology made the material interesting and highlighting the worse outcomes in high-risk patients leverages the importance of material from a clinical aspect of evaluation. The manuscript needs some revisions prior to finalization and reaching a decision. My comments and suggestions are provided below.

1. The first major concern about this manuscript is the used terms of “high-risk” and “very high-risk” patients with prostate cancer in need of surgery. The used nomenclature is not so usual in the literature in field. It is more questionable when the authors presented the results stratified by “below high-risk” and “high-risk and very high-risk”. Why not reporting the results for “low-risk” and “high-risk” patients as it is common in literature? A revision on this issue might be helpful.

2. Abstract: this section needs more information of the study methods and the definitions of study groups and outcomes in the materials and methods part, to make the following presented results clearer and easy to understand.

3. Introduction: the part on the review of previous literature and highlighting the gap in studies that is tried to be covered by this study is insufficient and needs more expansion and citation to similar previous studies.

4. Materials and Methods: The second paragraph introducing the two groups of <high and="" risk="">5. Statistical analysis: it should be clear whether the authors used “multivariate” or “multivariable” logistic regression models to investigate the predictive factors for postoperative outcomes, since these two are different methods statistically. Also, adding the log-rank test as the statistical test to compare the results of survival analysis is suggested in this part.

6. Results: some results in the tables are stratified by four risk groups and it was not mentioned in the methods what the criteria for this stratification was. One major issue about the results are the survival results presented in figure 1B. Since the provided P-value are between three recruited groups it is not clear whether the results between the high-risk groups with and without AT were significant or not. Adding another plot with only these two groups would be beneficial.

7. Discussion: although the results are compared with literature well, some studies are missed and it is suggested to included relevant studies about the results of RaRP in high-risk patients to this section (Example: Abdollah F, Sood A, Sammon JD, Hsu L, Beyer B, Moschini M, Gandaglia G, Rogers CG, Haese A, Montorsi F, Graefen M, Briganti A, Menon M. Long-term cancer control outcomes in patients with clinically high-risk prostate cancer treated with robot-assisted radical prostatectomy: results from a multi-institutional study of 1100 patients. Eur Urol. 2015 Sep;68(3):497-505. doi: 10.1016/j.eururo.2015.06.020. Epub 2015 Jun 26. PMID: 26119559.)</high>

Reviewer #2: Thanks for sending me this analytical work about prostate cancer robotic treatment. First of all congratulations to the authors for their work and achievement, I believe the aims of study are of utmost importance to the designated field but there are some major issues that need to be addressed before proceeding. Would be happy to help with revision and eventually see the paper published.

Majors:

1- The definition of risk (high, low, etc) are not defined while being used throughout the paper

2- The flow of paper (both scientific and narrative) is hard to follow. Several times I was lost in the middle of results and had to turn back and read from the beginning. In other words, the results section is not organized and is long. They may add some subsections

Minors:

1- Authors were too generous with using abbreviations. I strongly suggest using full wording as much as possible

2- I think it is MacOS Big Sur

Reviewer #3: In this paper, BCR-free survival rate as an oncological outcome and stress urinary incontinence (SUI) and Urgency Urinary Incontinence (UUI) as functional outcomes of robot assisted radical prostatectomy (RaRP) were investigated in high-risk and very high-risk prostate cancer patients and were compared to below high-risk group. The BCR-free survival of patients in high-risk and very high-risk group who also received adjuvant therapy was comparable to those in below high-risk group. Also, in long-term follow-up, the rate of SUI was not significantly different between high-risk/ very high-risk and low risk group. Suggesting that RaRP is a safe treatment option in high-risk/ very high-risk prostate cancer patients.

The paper is well-written and the content is interesting to the readers of the journal.

In my opinion, the paper could be accepted after making major/minor revisions as follows:

Major concerns

It is not clear how the patients were assessed in the selected time intervals. Was it through clinic visits?

It would be better to mention in the statistical analysis part that data was reported as mean ± SD and median (range).

As well as age, the relationship between other characteristics of the 71 patients included for SUI and UUI analysis should be mentioned.

In 5th paragraph of results: odds ratios should come with confidence intervals.

In tables 1, S1, and S2, the total columns need to be transferred before risk groups as the p-value is related to the comparison of the risk groups.

In table 2 and supplementary table 3 heading, the number of patients that were included and analyzed should be mentioned.

Heading of table 1 and S2 are similar, and their difference is unclear.

Figure 2: in the horizontal axis the interval between 0-1 months shouldn’t be equal to 1-3 months or 3-6 months.

The latest study that authors discussed and referred is for Jul 2021, there are some recent studies for example by Shin et al. on oncological and functional outcomes of RaRP that should be discussed here.

Minor concerns

For the reference [(11)] in the material and method part, [] is extra.

UUI is abbreviation for “Urge/ urgency urinary incontinence” not “urinary urgency incontinence”.

In 4th paragraph of results, after talking about 71 patients included, first the number (%) of patients in each group should be mentioned and then the age and other characteristics.

Style of reference 4 should be corrected. It’s a published paper and there is no need to mention the website.

6. PLOS authors have the option to publish the peer review history of their article (what does this mean?). If published, this will include your full peer review and any attached files.

Reviewer #1: **Yes: **Sina Azadnajafabad, MD, MPH

Reviewer #2: **Yes: **Esmaeil Mohammadi, MD MPH

Reviewer #3: No

---

## [Author Response · Author response to Decision Letter 0]

23 Jan 2023

Please refer yourself to the file "response to reviewer". We have addressed all the comments and revised our manuscript accordingly. Please refer to the revised manuscript. We also have highlighted all the revision. We have revised figure 1 and figure 2 and nearly all the tables. Please refer to the revised manuscript, figures and supplementary files. We are sincerely grateful for this opportunity.

---

## [Decision Letter · Decision Letter 1]

16 Feb 2023

Oncological and Functional Outcomes of High-Risk and Very High-Risk Prostate Cancer Patients after Robot-Assisted Radical Prostatectomy

PONE-D-22-25140R1

Dear Dr. Jiang,

We’re pleased to inform you that your manuscript has been judged scientifically suitable for publication and will be formally accepted for publication once it meets all outstanding technical requirements.

Kind regards,

Mohammad-Mahdi Rashidi, M.D.

Guest Editor

PLOS ONE

Additional Editor Comments (optional):

Reviewers' comments:

Reviewer's Responses to Questions

**Comments to the Author**

1. If the authors have adequately addressed your comments raised in a previous round of review and you feel that this manuscript is now acceptable for publication, you may indicate that here to bypass the “Comments to the Author” section, enter your conflict of interest statement in the “Confidential to Editor” section, and submit your "Accept" recommendation.

Reviewer #1: All comments have been addressed

Reviewer #2: All comments have been addressed

Reviewer #3: All comments have been addressed

2. Is the manuscript technically sound, and do the data support the conclusions?

Reviewer #1: Yes

Reviewer #2: Yes

Reviewer #3: Yes

3. Has the statistical analysis been performed appropriately and rigorously? 

Reviewer #1: Yes

Reviewer #2: Yes

Reviewer #3: Yes

4. Have the authors made all data underlying the findings in their manuscript fully available?

Reviewer #1: Yes

Reviewer #2: Yes

Reviewer #3: Yes

5. Is the manuscript presented in an intelligible fashion and written in standard English?

Reviewer #1: Yes

Reviewer #2: Yes

Reviewer #3: Yes

6. Review Comments to the Author

Reviewer #1: With many thanks, the authors have carefully addressed my comments and suggestions in this revision. I endorse this manuscript for publication in its current format.

Reviewer #2: Authors did a great job and revised the paper accordingly. I have no further suggestions at this point

Reviewer #3: (No Response)

7. PLOS authors have the option to publish the peer review history of their article (what does this mean?). If published, this will include your full peer review and any attached files.

Reviewer #1: **Yes: **Sina Azadnajafabad, MD, MPH

Reviewer #2: **Yes: **Esmaeil Mohammadi, MD MPH

Reviewer #3: **Yes: **Niloufar Salehi

---

## [Editor Report · Acceptance letter]

21 Feb 2023

PONE-D-22-25140R1 

Oncological and Functional Outcomes of High-Risk and Very High-Risk Prostate Cancer Patients after Robot-Assisted Radical Prostatectomy 

Dear Dr. Jiang:

I'm pleased to inform you that your manuscript has been deemed suitable for publication in PLOS ONE. Congratulations! Your manuscript is now with our production department. 

Kind regards, 

on behalf of

Dr. Mohammad-Mahdi Rashidi 

Guest Editor

PLOS ONE